

# Prognostic value of mean platelet volume/platelet count ratio in patients with resectable esophageal squamous cell carcinoma: a retrospective study

Ji-Feng Feng[1,2], Chen Sheng[1], Qiang Zhao[1,2] and Pengcheng Chen[1]

[1] Department of Thoracic Surgery, Zhejiang Cancer Hospital, Hangzhou, China
[2] Key Laboratory Diagnosis and Treatment Technology on Thoracic Oncology, Hangzhou, China

## ABSTRACT

**Background**. Mean platelet volume (MPV) to platelet count (PC) ratio (MPV/PC) is a useful indicator in several cancers. However, the role for MPV/PC ratio in esophageal squamous cell carcinoma (ESCC) is still controversial.

**Methods**. A retrospective study was conducted including 277 resectable ESCC patients. The optimal cut-off values were calculated by the X-tile program. The receiver operator characteristic (ROC) curves were also created to show the candidate cut-off points. The comparisons between the X-tile plot and ROC curve were performed. The Kaplan-Meier method was utilized to analyze the cancer-specific survival (CSS). Prognostic factors for CSS were calculated with Cox regression univariate and multivariate analyses.

**Results**. According to the X-tile program, the cut-off values for MPV, PC and MPV/PC ratio were 8.5 (fl), 200 (giga/l) and 0.04, respectively. However, the cut-off values for MPV, PC and MPV/PC ratio by the ROC curves were 8.25 (fl), 243.5 (giga/l) and 0.0410, respectively. The cut-off values were similar between the X-tile and ROC curve. A low MPV/PC ratio level ($\leq$0.04) was associated with poor CSS (22.4% vs. 43.1%, $P < 0.001$). In multivariate analyses, we found that MPV/PC ratio was an independent predictor for CSS ($P < 0.001$). When we set the cut-off point using ROC curve, the MPV/PC ratio was still an independent predictor for CSS ($P < 0.001$).

**Conclusion**. The MPV/PC ratio is a useful predictive indicator in patients with ESCC.

Corresponding authors
Ji-Feng Feng, Fengjf@zjcc.org.cn
Pengcheng Chen,
Chenpc0425@126.com

## INTRODUCTION

Esophageal cancer (EC) is the 8th most common cancer worldwide and the 6th most common cause of death from cancer (*Ferlay et al., 2010*). The incidences vary widely in different countries and regions. To date, approximately 53.8% and 51.9% of all ECs occurred and died in China (*Siegel, Miller & Jemal, 2015*; *Ferlay et al., 2010*). Esophageal adenocarcinoma is the most common malignancy in the West. In China, however, esophageal squamous cell carcinoma (ESCC) is the predominant subtype (*Napier, Scheerer & Misra, 2014*). Radical esophagectomy remains the most effective therapy for patients with EC. However, the prognosis for EC remains poor (*Bedenne et al., 2007*; *Domper Arnal,*

*Ferrández Arenas & Lanas Arbeloa, 2015*). Therefore, it is very important to find more and more useful and effective prognostic indicators for patients with EC.

Over the past few decades, a number of prognostic factors for EC have been identified, including tumor length, vessel invasion, lymph node status (N stage), depth of invasion (T stage), TNM stage and other serum biomarkers, such as squamus cell carcinoma antigen (SCCA) and carcinoembryonic antigen (CEA) (*Peyre et al., 2008*; *Wijnhoven et al., 2007*; *Feng, Huang & Zhao, 2013*). Inflammation plays an important role in cancer progression and prognosis (*Balkwill & Mantovani, 2001*; *Mantovani et al., 2008*). C-reactive protein (CRP), as a most sensitive inflammatory biomarker, has been confirmed in a series of cancers to predict the prognosis, including patients with EC (*Shimada et al., 2003*; *Nozoe, Saeki & Sugimachi, 2001*; *Platt et al., 2012*). In addition, there are other parameters like neutrophil and lymphocyte that are easy-to-measure inflammatory markers (*Dutta et al., 2011*).

Mean platelet volume (MPV) is recognized as a hallmark for platelet count (PC) activation (*Kamath, Blann & Lip, 2001*). Several studies showed that MPV and PC are associated with mortality in cardiovascular disease, such as ischemic cardiovascular disease and acute myocardial infarction (*Guenancia et al., 2017*; *Azab et al., 2011*). Moreover, recent studies have shown that the ratio for MPV to PC (MPV/PC) is associated with prognosis in some malignancies, such as hepatocellular carcinoma and lung cancer (*Cho et al., 2013*; *Inagaki et al., 2014*; *Omar et al., 2018*). However, the role for MPV/PC ratio in ESCC is still controversial. Furthermore, controversy exists concerning the optimal cut-off points for MPV/PC to predict the prognosis of ESCC. Therefore, the purpose of our study here was to explore the prognostic role of MPV/PC ratio in patients with ESCC.

## PATIENTS AND METHODS

From January 2007 to December 2010 at the Department of Thoracic Surgery, Zhejiang Cancer Hospital, a retrospective study was conducted including 277 resectable ESCC patients. The exclusion criteria were as follows: (1) patients who received preoperative treatment, such as chemotherapy and/or radiotherapy; (2) patients who had any form of acute or chronic inflammatory diseases or infections; (3) patients who had systemic diseases, and (4) those diagnosed with distant metastases. Written informed consent for the collection of specimen and other medical information were obtained from all patients before surgery. The current study was approved by the Ethics Committees of Zhejiang Cancer Hospital (IRB Approval No. IRB-2018-130).

The main clinical characteristics, such as age, gender, tumor location (upper, middle and lower), tumor length, vessel invasion, differentiation (well, moderate and poor) and tumor stage (T stage, N stage and TNM stage), were retrospectively reviewed and collected. The tumor length was defined as the long diameter for pathological specimens. Blood samples were obtained within one week prior to surgery to measure the neutrophil (Neu), MPV, PC, CRP and CEA levels. MPV/PC was defined as MPV to PC ratio. Neu/PC was defined as Neu to PC ratio. The levels of Neu, MPV and PC were measured by automated blood cell counter (Sysmex XE-2100; Sysmex, Kobe, Japan). Serum levels of CRP were

determined by latex-enhanced homogeneous immunoassay (Hitachi 917; Skill, Munich, Germany). Serum levels of CEA were measured using enzyme immunoassay kits (Abbott, Chicago, IL, USA). The AJCC/UICC TNM staging system (the 7th edition) was utilized to classify the stage for this study (*Rice et al., 2010*).

All the above patients were followed-up postoperatively (regularly evaluated every 3–6 months). The assessment included physical examination, blood tumor markers and computed tomography scan. In this study, we conducted a cancer-specific survival (CSS) to analyze the prognosis of patients with ESCC. The mean follow-up for patients was 45 months.

## Statistical analyses

In the current study, the optimal cut-off values for Neu, MPV, PC, MPV/PC ratio, and Neu/PC ratio were calculated by the X-tile program (*Camp, Dolled-Filhart & Rimm, 2004*). The receiver operator characteristic (ROC) curves were also created to show the candidate cut-off points. The comparisons between the X-tile plot and ROC curve were performed. The areas under the curve (AUC) for Neu, MPV, PC, MPV/PC and Neu/PC were calculated and compared by the ROC curve. The chi-squared tests were used to compare the MPV/PC ratio, MPV and PC. The CSS curves were generated by the Kaplan–Meier method. Univariate analyses were performed with log-rank test. Multivariate analyses with cox proportional hazards regression model were utilized to analyze prognostic factors for CSS. SPSS 20.0 (SPSS Inc., Chicago, IL, USA) was utilized to perform the statistical analyses. R 3.2.3 software (Institute for Statistics and Mathematics, Vienna, Austria) was also utilized to conduct the nomogram model by Harrell's concordance index (c-index) (*Iasonos et al., 2008*).

## RESULTS

There were 37 (13.4%) women and 240 (86.6%) men in all 277 patients with the mean age of $59.2 \pm 7.8$ years (36-80 years). In the current study, the mean values for Neu, MPV, PC, MPV/PC and Neu/PC were $4.2 \pm 1.5$ (giga/l) (range 1.5–9.5 giga/l), $9.3 \pm 1.3$ (fl) (range 6.7–12.9 fl), $232 \pm 72$ (giga/l) (range 60-473 giga/l), $0.04 \pm 0.02$ (range 0.02–0.14), and $0.020 \pm 0.010$ (range 0.0053–0.0667), respectively. The histograms of the preoperative MPV/PC ratio, MPV and PC are shown in Fig. 1.

According to the X-tile program, the optimum cut-off points for MPV, PC, MPV/PC, Neu and Neu/PC ratio were 8.5 (fl), 200 (giga/l), 0.04, 4.2 (giga/l) and 0.02, respectively (Fig. 2). According to the optimum cut-off points of the above values, patients then were divided into 2 groups (MPV ≤8.5 fl and >8.5 fl; PC ≤200 giga/l and >200 giga/l; MPV/PC ratio ≤0.04 and >0.04). Clinicopathologic characters for the above values (MPV/PC ratio, MPV and PC) were shown in Table 1. The levels of MPV/PC ratio were significantly correlated with the CRP levels ($P = 0.029$).

Kaplan–Meier analyses showed that a low MPV/PC ratio level (≤0.04) was associated with poor CSS ($P < 0.001$). The 5-year CSS was 43.1% in patients with MPV/PC ratio >0.04, and 22.4% in patients with MPV/PC ratio ≤0.04 (Fig. 3A). There were also significantly different for MPV (42.4% vs. 27.0%, $P = 0.010$) and PC (41.0% vs. 26.7%,

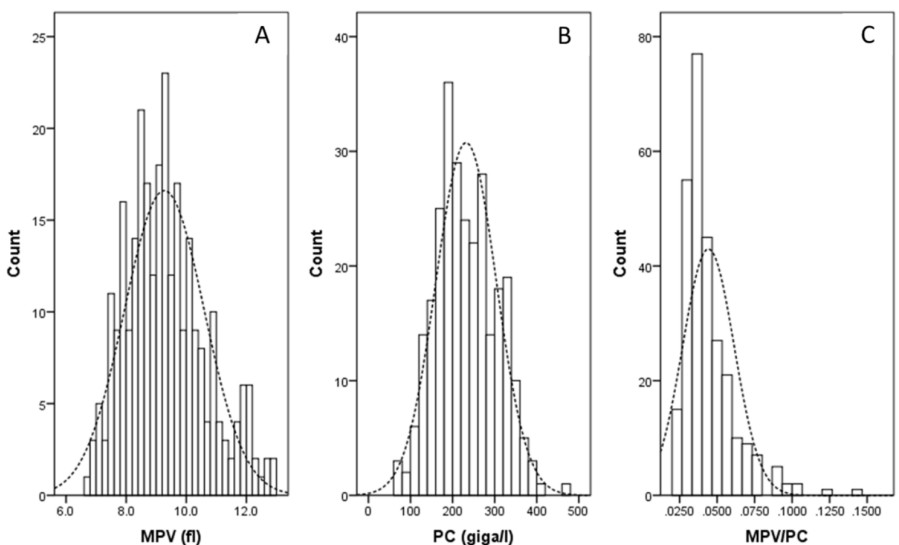

**Figure 1** The histograms of the MPV (A), PC (B) and MPV/PC ratio (C).

$P = 0.009$) (Figs. 3B–3C). In multivariate analyses, we found that MPV/PC ratio was an independent predictor for CSS ($P < 0.001$) (Table 2). In addition, TNM stage ($P < 0.001$), CEA ($P = 0.019$), Neu ($P = 0.007$) and CRP ($P < 0.001$) were other significant prognostic variables by multivariate analyses (Table 2).

We also created ROC curves to show the candidate cut-off points. The cut-off values for Neu, MPV, PC, MPV/PC, and Neu/PC ratio by the ROC curves were 4.25 (giga/l), 8.25(fl), 243.5 (giga/l), 0.0410, and 0.0213, respectively (Fig. 4). The candidate cut-off points and the area under ROC curve (AUC) are shown in Table 3. When we set the cut-off points using ROC curve, the MPV/PC ratio (42.7% vs. 23.5%, $P < 0.001$), MPV (51.7% vs. 26.7%, $P = 0.001$), and PC (41.8% vs. 19.3%, $P < 0.001$) were also associated with CSS (D-F) (Figs. 3D–3F). In multivariate analyses, MPV/PC ratio was still an independent predictor for CSS ($P < 0.001$) (Table 4).

Moreover, we wanted to predict the survival risk (CSS) for patients with ESCC, a nomogram model was conducted including age, gender, TNM, CEA, Neu, MPV/PC ratio and CRP for CSS (Fig. 5). From this model, the probability of survival for ESCC patients could be predicted (c-index = 0.72).

## DISCUSSION

Our study demonstrated some important findings: (1) MPV/PC ratio was a strong predictor of CSS; (2) MPV/PC ratio, but not MPV or PC, was a useful predictive indicator. This study used X-tile program and ROC curves as candidate cut-off points. The comparisons between the X-tile plot and ROC curve were performed. The cut-off values were similar between the X-tile and ROC curve. Moreover, our study is also the first attempt to predict the survival risk by a nomogram model based on MPV/PC ratio.

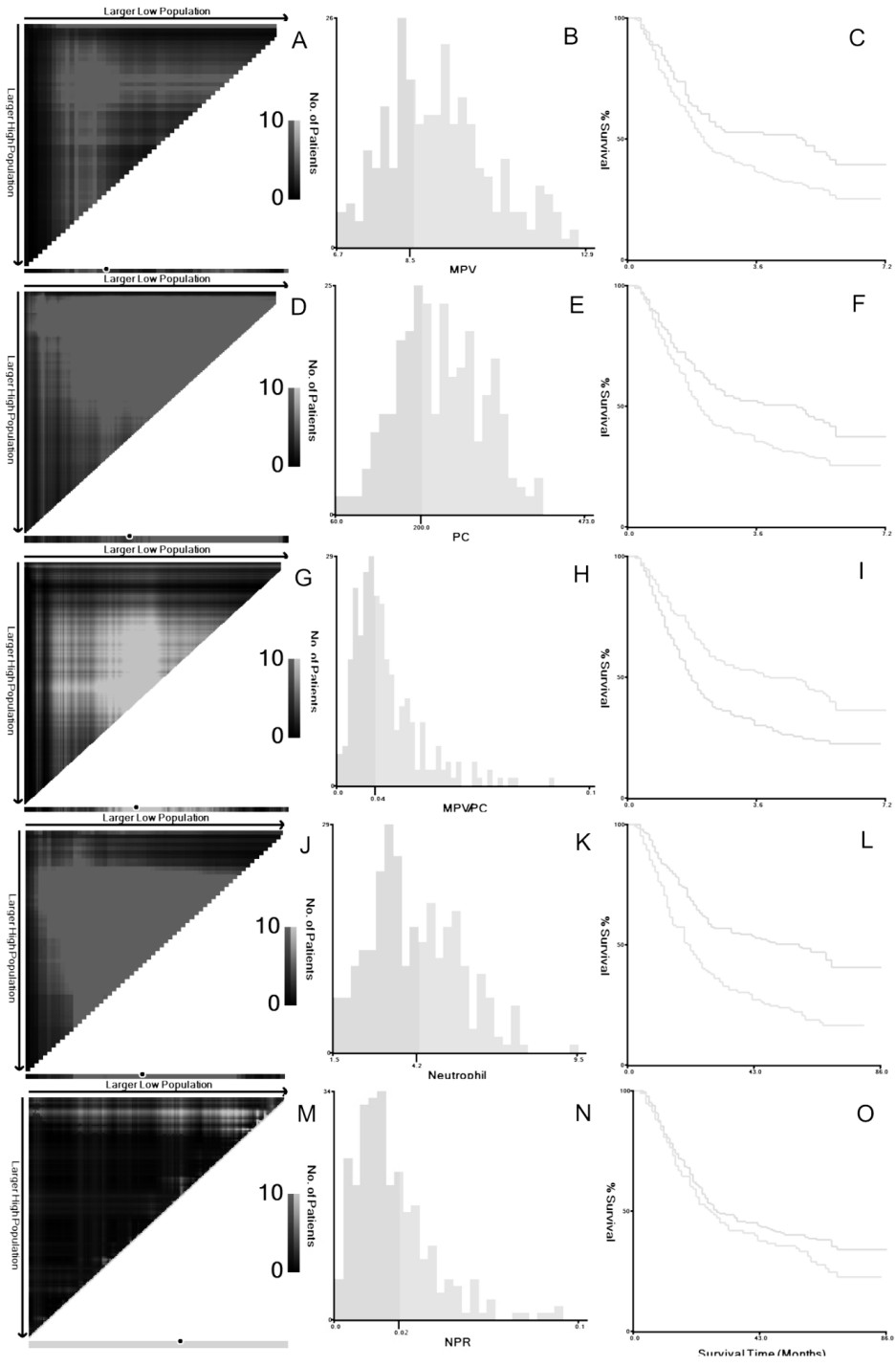

**Figure 2  X-tile analyses.** X-tile plots of the training sets are shown in A, D, G, J, M, with plots of matched validation sets shown in the smaller inset. The optimal cut-off points highlighted by the black circle in A, D, G, J and M are shown on the histograms of the entire cohort (B, E, H, K, N) and Kaplan-Meier plots (C, F, I, L, O). According to the X-tile program, the optimum cut-off points for MPV (A–C), PC (D–F), MPV/PC (G–I), Neu (J–L) and Neu/PC ratio (M–O) were 8.5 (fl), 200 (giga/l), 0.04, 4.2 (giga/l) and 0.02, respectively.

**Table 1  Comparison of baseline clinical characteristics in ESCC.**

| | Total | MPV (fl) | | P value | PC (giga/l) | | P value | MPV/PC | | P value |
|---|---|---|---|---|---|---|---|---|---|---|
| | | ≤8.5 | >8.5 | | ≤200 | >200 | | ≤0.04 | >0.04 | |
| Age (years) | | | | 0.704 | | | 0.221 | | | 0.488 |
| ≤60 | 158 | 51 | 107 | | 55 | 103 | | 81 | 77 | |
| >60 | 119 | 41 | 78 | | 50 | 69 | | 66 | 53 | |
| Gender | | | | 0.521 | | | 0.271 | | | 0.629 |
| Female | 37 | 14 | 23 | | 11 | 26 | | 21 | 16 | |
| Male | 240 | 78 | 162 | | 94 | 146 | | 126 | 114 | |
| Tumor length (cm) | | | | 0.246 | | | 0.020 | | | 0.087 |
| ≤3.0 | 78 | 30 | 48 | | 38 | 40 | | 35 | 43 | |
| >3.0 | 199 | 62 | 137 | | 67 | 132 | | 112 | 87 | |
| CRP (mg/l) | | | | 0.031 | | | 0.152 | | | 0.029 |
| ≤10.0 | 200 | 74 | 126 | | 81 | 119 | | 98 | 102 | |
| >10.0 | 77 | 18 | 59 | | 24 | 53 | | 49 | 28 | |
| Tumor location | | | | 0.242 | | | 0.096 | | | 0.057 |
| Upper | 16 | 7 | 9 | | 10 | 6 | | 4 | 12 | |
| Middle | 127 | 36 | 91 | | 44 | 83 | | 72 | 55 | |
| Lower | 134 | 49 | 85 | | 51 | 83 | | 71 | 63 | |
| Vessel invasion | | | | 0.744 | | | 0.097 | | | 0.111 |
| Negative | 232 | 78 | 154 | | 83 | 149 | | 128 | 104 | |
| Positive | 45 | 14 | 31 | | 22 | 23 | | 19 | 26 | |
| Differentiation | | | | 0.927 | | | 0.826 | | | 0.454 |
| Well | 43 | 15 | 28 | | 16 | 27 | | 25 | 18 | |
| Moderate | 179 | 58 | 121 | | 70 | 109 | | 90 | 89 | |
| Poor | 55 | 19 | 36 | | 19 | 36 | | 32 | 23 | |
| T stage | | | | 0.106 | | | 0.313 | | | 0.425 |
| T1 | 50 | 22 | 28 | | 18 | 32 | | 28 | 22 | |
| T2 | 49 | 20 | 29 | | 24 | 25 | | 21 | 28 | |
| T3 | 154 | 44 | 110 | | 56 | 98 | | 86 | 68 | |
| T4 | 24 | 6 | 18 | | 7 | 17 | | 12 | 12 | |
| N stage | | | | 0.054 | | | 0.720 | | | 0.899 |
| N0 | 150 | 60 | 90 | | 61 | 89 | | 78 | 72 | |
| N1 | 74 | 21 | 53 | | 27 | 47 | | 39 | 35 | |
| N2 | 32 | 7 | 25 | | 10 | 22 | | 19 | 13 | |
| N3 | 21 | 4 | 17 | | 7 | 14 | | 11 | 10 | |
| TNM stage | | | | 0.003 | | | 0.357 | | | 0.546 |
| I | 69 | 31 | 38 | | 31 | 38 | | 33 | 36 | |
| II | 92 | 35 | 57 | | 34 | 58 | | 52 | 40 | |
| III | 116 | 26 | 90 | | 40 | 76 | | 62 | 54 | |
| CEA (ng/ml) | | | | 0.818 | | | 0.566 | | | 0.954 |
| ≤5.0 | 239 | 80 | 159 | | 89 | 50 | | 127 | 112 | |
| >5.0 | 38 | 12 | 26 | | 16 | 22 | | 20 | 18 | |

| | Total | MPV (fl) | | *P* value | PC (giga/l) | | *P* value | MPV/PC | | *P* value |
| --- | --- | --- | --- | --- | --- | --- | --- | --- | --- | --- |
| | | ≤8.5 | >8.5 | | ≤200 | >200 | | ≤0.04 | >0.04 | |
| Neu (giga/l) | | | | 0.249 | | | 0.681 | | | 0.186 |
| ≤4.2 | 146 | 53 | 93 | | 57 | 89 | | 72 | 74 | |
| >4.2 | 131 | 39 | 92 | | 48 | 83 | | 75 | 56 | |
| Neu/PC | | | | 0.090 | | | <0.001 | | | <0.001 |
| ≤0.02 | 170 | 50 | 120 | | 35 | 135 | | 114 | 56 | |
| >0.02 | 107 | 42 | 65 | | 70 | 37 | | 33 | 74 | |

**Notes.**

ESCC, esophageal squamous cell carcinoma; CRP, c-reactive protein; MPV, mean platelet volume; PC, platelet count; TNM, tumor node metastasis; CEA, carcinoembryonic antigen; Neu, neutrophil.

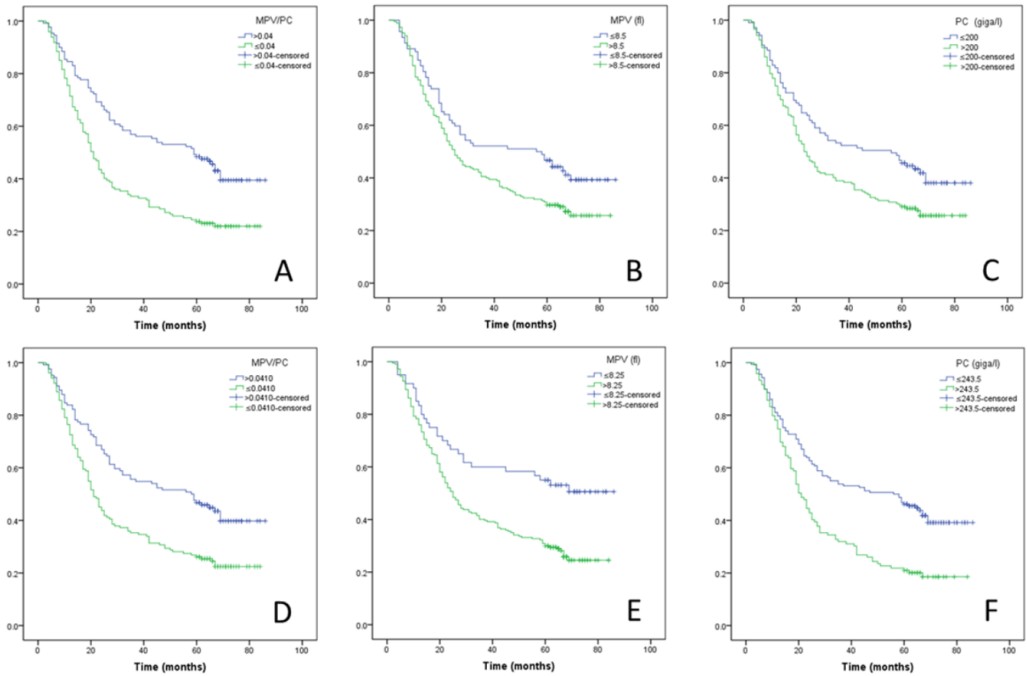

**Figure 3 Kaplan–Meier CSS curves.** Patients with MPV/PC ratio > 0.04 had a significantly better 5-year CSS than patients with MPV/PC ratio ≤ 0.04 (43.1% vs. 22.4%, $P < 0.001$; (A). The 5-year CSS were also significantly different for MPV (42.4% vs. 27.0%, $P = 0.010$; (B) and PC (41.0% vs. 26.7%, $P = 0.009$; (C). When we set the cut-off points using ROC curve, the MPV/PC ratio (42.7% vs. 23.5%, $P < 0.001$; (D), MPV (51.7% vs. 26.7%, $P = 0.001$; (E), and PC (41.8% vs. 19.3%, $P < 0.001$; (F) were also associated with CSS.

Platelet activation has been demonstrated as a common phenomenon in some cardiovascular diseases (*Guenancia et al., 2017*; *Azab et al., 2011*). To assess the platelet activation status, MPV and PC are two main aspects. Moreover, studies have shown that MPV/PC ratio is associated with prognosis in some malignancies, such as hepatocellular carcinoma and lung cancer (*Cho et al., 2013*; *Inagaki et al., 2014*; *Omar et al., 2018*). *Cho et al. (2013)* have shown that the ratio of MPV/PC levels in hepatocellular carcinoma were higher than the control group. *Inagaki et al. (2014)* have revealed that MPV/PC ratio was

**Table 2  Univariate and multivariate analyses for cancer-specific survival.**

| | CSS (%) | *P* value | Univariate analysis HR (95% CI) | *P* value | Multivariate analysis HR (95% CI) | *P* value |
|---|---|---|---|---|---|---|
| Age (years) | | 0.412 | | 0.417 | – | – |
| ≤60 | 33.5 | | 1.000 | | | |
| >60 | 30.3 | | 1.127 (0.845–1.502) | | | |
| Gender | | 0.114 | | 0.120 | – | – |
| Female | 45.9 | | 1.000 | | | |
| Male | 30.0 | | 1.445 (0.909–2.298) | | | |
| Tumor length | | 0.003 | | 0.004 | – | – |
| ≤3 cm | 42.3 | | 1.000 | | | |
| > 3 cm | 28.1 | | 1.642 (1.173–2.297) | | | |
| Tumor location | | 0.336 | | 0.342 | – | – |
| Upper/Middle | 35.7 | | 1.000 | | | |
| Lower | 28.4 | | 1.149 (0.863–1.530) | | | |
| Vessel invasion | | 0.003 | | 0.003 | – | – |
| Negative | 35.3 | | 1.000 | | | |
| Positive | 15.6 | | 1.710 (1.197–2.444) | | | |
| Differentiation | | 0.054 | | 0.058 | – | – |
| Well/Moderate | 33.8 | | 1.000 | | | |
| Poor | 25.5 | | 1.398 (0.989–1.978) | | | |
| T stage | | <0.001 | | <0.001 | – | – |
| T1-2 | 45.5 | | 1.000 | | | |
| T3-4 | 24.7 | | 1.898 (1.382–2.606) | | | |
| N stage | | <0.001 | | <0.001 | – | – |
| N0 | 49.3 | | 1.000 | | | |
| N1-3 | 11.8 | | 2.852 (2.120–3.836) | | | |
| TNM stage | | <0.001 | | <0.001 | | <0.001 |
| I | 58.0 | | 1.000 | | 1.000 | |
| II | 38.9 | | 1.966 (1.259–3.067) | 0.003 | 1.825 (1.164–2.861) | 0.009 |
| III | 13.8 | | 3.799 (2.490–5.736) | <0.001 | 3.624 (2.362–5.560) | <0.001 |
| Adjuvant therapy | | 0.085 | | 0.090 | – | – |
| No | 35.6 | | 1.000 | | | |
| Yes | 24.4 | | 1.297 (0.960–1.753) | | | |
| CRP (mg/l) | | <0.001 | | <0.001 | | <0.001 |
| ≤ 10.0 | 39.5 | | 1.000 | | 1.000 | |
| > 10.0 | 13.0 | | 2.066 (1.526–2.798) | | 1.994 (1.461–2.722) | |

**Table 2** (*continued*)

|  | CSS (%) | *P* value | Univariate analysis HR (95% CI) | *P* value | Multivariate analysis HR (95% CI) | *P* value |
|---|---|---|---|---|---|---|
| MPV (fl) |  | 0.019 |  | 0.021 | – | – |
| ≤ 8.5 | 41.3 |  | 1.000 |  |  |  |
| > 8.5 | 27.6 |  | 1.451 (1.057–1.992) |  |  |  |
| PC (giga/l) |  | 0.009 |  | 0.011 | - | - |
| ≤ 200 | 41.0 |  | 1.000 |  |  |  |
| > 200 | 26.7 |  | 1.488 (1.097–2.019) |  |  |  |
| MPV/PC |  | <0.001 |  | <0.001 |  | <0.001 |
| > 0.04 | 43.1 |  | 1.000 |  | 1.000 |  |
| ≤ 0.04 | 22.4 |  | 1.861 (1.386–2.498) |  | 1.823 (1.347–2.469) |  |
| CEA (ng/ml) |  | 0.027 |  | 0.031 |  | 0.019 |
| ≤ 5.0 | 33.5 |  | 1.000 |  | 1.000 |  |
| > 5.0 | 23.7 |  | 1.549 (1.042–2.302) |  | 1.613 (1.082–2.407) |  |
| Neu (giga/l) |  | <0.001 |  | <0.001 |  | 0.007 |
| ≤ 4.2 | 43.8 |  | 1.000 |  | 1.000 |  |
| > 4.2 | 19.1 |  | 1.945 (1.455–2.600) |  | 1.512 (1.120–2.040) |  |
| Neu/PC |  | 0.223 |  | 0.229 | – | – |
| ≤ 0.02 | 35.3 |  | 1.000 |  |  |  |
| > 0.02 | 27.1 |  | 1.195 (0.894–1.597) |  |  |  |

**Notes.**
ESCC, esophageal squamous cell carcinoma; CRP, c-reactive protein; MPV, mean platelet volume; PC, platelet count; TNM, tumor node metastasis; CEA, carcinoembryonic antigen; Neu, neutrophil; CI, confidence interval; HR, hazard ratio.

**Table 3  Comparison of AUC areas for the prognostic factors in ESCC.**

|  | Cut-off | Sensibility | Specificity | AUC | 95% CI | *P*-value |
|---|---|---|---|---|---|---|
| MPV/PC | 0.0410 | 62.9 | 61.7 | 0.608 | 0.548–0.666 | Reference |
| MPV | 8.25 | 84.6 | 34.8 | 0.609 | 0.549–0.667 | 0.9834 |
| PC | 243.5 | 50.0 | 74.2 | 0.648 | 0.588–0.704 | 0.0181 |
| Neu | 4.25 | 53.7 | 76.4 | 0.689 | 0.630–0.743 | 0.1123 |
| Neu/PC | 0.0213 | 38.3 | 74.2 | 0.543 | 0.482–0.603 | 0.3269 |

**Notes.**
ESCC, esophageal squamous cell carcinoma; AUC, area under the curve; MPV, mean platelet volume; PC, platelet count; Neu, neutrophil.

significantly different on survival in lung cancer. However, *Omar et al. (2018)* showed that increased MPV and increased PC were significant higher than the control group. In their study, however, MPV/PC was not an independent predictor in lung cancer.

MPV is an indicator of platelet activation. *Shen et al. (2018)* demonstrated that reduced MPV is associated with worse survival outcome in EC. The role for MPV/PC ratio in ESCC patients has not yet been well evaluated. A study reported by *Sun et al. (2018)* showed that the levels of MPV/PC ratio in ESCC were significantly lower than the healthy group, and which were significantly correlated with the tumor length. In our study, however, the MPV/PC ratio was not significantly correlated with the tumor length ($P = 0.087$). In addition, they revealed that decreased MPV and MPV/PC ratio were significantly

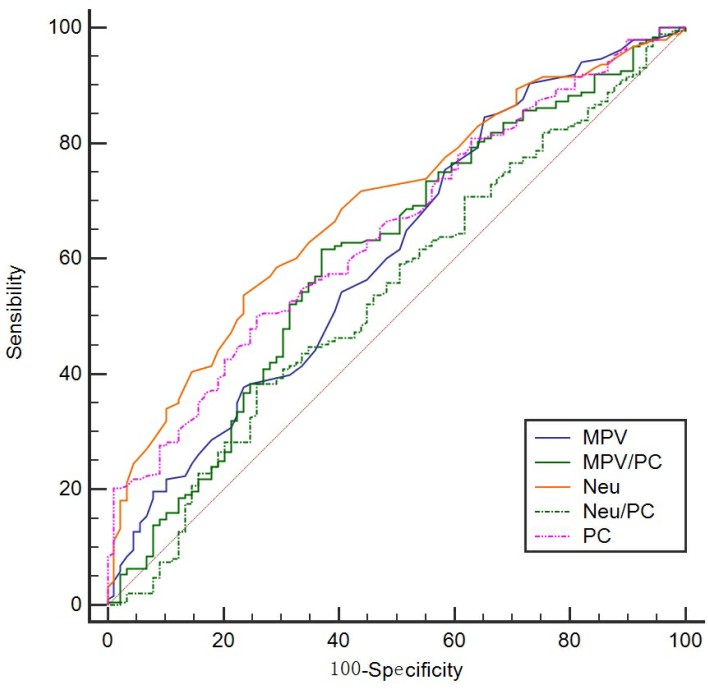

**Figure 4  ROC curve analysis.** The cut-off values for Neu, MPV, PC, MPV/PC, and Neu/PC ratio by the ROC curves were 4.25 (giga/l), 8.25(fl), 243.5 (giga/l), 0.0410, and 0.0213, respectively.

**Table 4  Multivariate analyses in ESCC with the cut-off values by ROC curve.**

|  | HR (95% CI) | *P*-value |
|---|---|---|
| CRP (mg/l) (>10.0 vs. ≤10.0) | 2.060 (1.511–2.807) | <0.001 |
| TNM stage |  |  |
|     II vs. I | 1.816 (1.160–2.844) | 0.009 |
|     III vs. I | 3.529 (2.298–5.417) | <0.001 |
| MPV/PC ( ≤0.0410 vs. >0.0410) | 1.728 (1.275–2.342) | <0.001 |
| CEA (ng/ml) ( >5.0 vs. ≤5.0) | 1.636 (1.097–2.438) | 0.016 |
| Neu (giga/l) ( >4.25 vs. ≤4.25) | 1.553 (1.150–2.096) | 0.004 |

**Notes.**
ESCC, esophageal squamous cell carcinoma; CSS, cancer-specific survival; CRP, c-reactive protein; MPV, mean platelet volume; PC, platelet count; TNM, tumor node metastasis; CEA, carcinoembryonic antigen; Neu, neutrophil; CI, confidence interval; HR, hazard ratio.

associated with locally advanced ESCC. In our study, MPV was not a significant prognostic factor by multivariate analyses. Recently, *Zhang et al. (2016)* initial conducted a COP-MPV (combination of MPV and PC) model to predict the prognosis in ESCC. They revealed that COP-MPV was a useful independent predictor, but not for MPV or PC. As everyone knows, MPV and PC may be influenced by a variety of other non-cancer related conditions, the potential basis could be decreased by the MPV to PC ratio (MPV/PC). Therefore, the role of the MPV/PC ratio would be more reliable than the effect of either MPV or PC. In

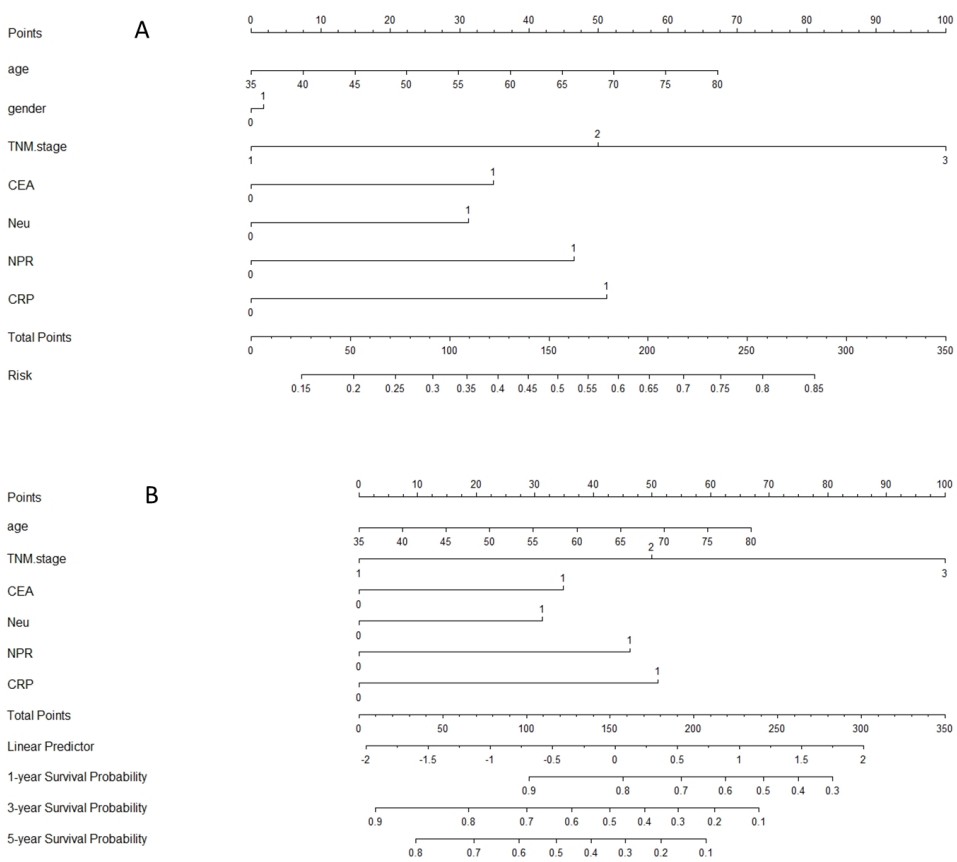

**Figure 5  Nomogram model for prediction.** The Harrell's c-index for CSS prediction was 0.72. A nomogram predicts survival prediction based on MPV/PC and other prognostic factors in patients with ESCC. The nomogram is used by totalling the points identified at the top of the scale for each independent factor. This total point score is then identified on the total points scale to determine the probability of risk prediction (A) and survival prediction (B).

the current study, a low MPV/PC ratio level ($\leq 0.04$) was associated with poor CSS ($P < 0.001$) and was confirmed by multivariate analyses ($P < 0.001$).

In previous studies, controversy exists about the optimum cut-off point for MPV/PC ratio to predict prognosis. *Cho et al. (2013)* demonstrated that 0.0491 might be the optimum cut-off point for MPV/PC ratio in hepatocellular carcinoma according to the ROC curve. *Inagaki et al. (2014)* and *Omar et al. (2018)* also conducted the ROC curve analyses to calculate the optimum cut-off point for MPV/PC in lung cancer. They concluded that the optimum cut-off points for MPV/PC ratio were 0.40873 and 0.47424, respectively. Recently, *Camp, Dolled-Filhart & Rimm (2004)* initial conducted a program to explore the optimum cut-off point (X-tile plot). In our study, according to their method, 0.04 was the optimum cut-off point for MPV/PC ratio. We also created ROC curves to show the candidate cut-off points. When we set the cut-off point using ROC curve, the MPV/PC ratio was also associated with CSS. In multivariate analyses, MPV/PC ratio was still an independent predictor for CSS.

The mechanism between MPV/PC ratio and cancer remains unknown. Inflammation and cancer are closely related (*Balkwill & Mantovani, 2001*; *Mantovani et al., 2008*). As is well known, platelets can release a variety of cytokines, such as platelet-derived growth factor (PDGF) and vascular endothelial growth factor (VEGF), which have an important role in regulating angiogenesis (*Blair & Flaumenhaft, 2009*; *Borsig, 2008*; *Dineen et al., 2009*). The inflammation will be inevitably caused by chemotherapy and/or radiation. Therefore, we analyze the role of MPV/PC ratio in ESCC patients without neoadjuvant chemotherapy and/or radiation.

Limitations should be acknowledged in this study. The major limitations of this study are small samples and its retrospective character. Moreover, patients who received preoperative chemotherapy and/or radiotherapy were excluded, which might have influenced the result in the current study. On the one hand, neoadjuvant treatment will have a side effect on MPV and PC. On the other hand, neoadjuvant treatment can improve cancer survival for locally advanced EC, but not for early stage EC (*Rawat et al., 2013*; *Mariette et al., 2014*). In addition, we did not set up a validation group to verify the conclusion. Thus, the results of our study are expected more large-sample trials to confirm in future.

## CONCLUSION

In summary, we found that the ratio of MPV/PC is a potential prognostic biomarkers in patients with ESCC.

### Funding
This work was supported by the Medical Health Science and Technology Project of Zhejiang Provincial Health Commission (No. 2018KY290, 2019RC129). The funders had no role in study design, data collection and analysis, decision to publish, or preparation of the manuscript.

### Grant Disclosures
The following grant information was disclosed by the authors:
Medical Health Science and Technology Project of Zhejiang Provincial Health Commission: 2018KY290, 2019RC129.

### Competing Interests
The authors declare there are no competing interests.

### Author Contributions
- Ji-Feng Feng conceived and designed the experiments, performed the experiments, contributed reagents/materials/analysis tools, authored or reviewed drafts of the paper, approved the final draft.
- Chen Sheng and Qiang Zhao analyzed the data, prepared figures and/or tables, authored or reviewed drafts of the paper, approved the final draft.

## PeerJ

- Pengcheng Chen performed the experiments, authored or reviewed drafts of the paper, approved the final draft.

## Human Ethics

The following information was supplied relating to ethical approvals (i.e., approving body and any reference numbers):

The current study was approved by the Ethics Committees of Zhejiang Cancer Hospital (Approval No. IRB-2018-130).

## Data Availability

The raw measurements are available in the Supplemental Files.

## Supplemental Information

Supplemental information for this article can be found online at http://dx.doi.org/10.7717/peerj.7246#supplemental-information.

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
