# Peer review of "Prognostic value of mean platelet volume/platelet count ratio in patients with resectable esophageal squamous cell carcinoma: a retrospective study"

_PeerJ, doi:10.7717/peerj.7246_

## Round 0.1 · original submission · Major Revisions

Please provide a point-to-point response to the insightful reviewer comments in your revision, as well as a tracked-changes copy.

[]

Reviewer 1 ·

Basic reporting

This is a retrospective study that can clearly illustrate the prognostic value of MPV/PC for esophageal squamous cell carcinoma. However, for this study, the sample size is not large enough, and the selected statistical methods appear still questionable.

Experimental design

1. Did the researchers set up a validation group to verify the conclusion?
2. Line 121-122, The levels of MPV/PC ratio were significantly correlated with the CRP levels (P =0.029). CRP is an inflammatory marker just like MPV/PC. These two markers would interact with each other in the Cox regression model to influence the final conclusion.

Validity of the findings

1. CEA is closely related to the occurrence, recurrence and metastasis of ESCC. Why didn’t researchers consider CEA as one of the prognostic factors of CSS?
2. Why did the researchers use the X-tile program instead of the ROC to calculate the optimal cut-off values? What are the advantages of this method relative to the ROC?
3. There are some grammatical errors/writing style issues. e.g in the abstract, “respectively” is needed at the end of the first sentence in the results section.

·

Basic reporting

İntroduction is too short and literature review is not satisfactory.

Experimental design

Feng et. al used X-tile program to find the cut-off vaules. I prefer to find these valueas with ROC analyses. they should make the statistics again. The results are not consşistent with the other reports in the literature. How do the author explain that MPV is not an independent prognostic factor but MPV/PC is a prognostic factor. İs it a numerical chance?

Validity of the findings

statistical analyses are not satisfactory and results are not consistent with the literature. I dont know the spesifity and sensitivity of the cut -off values.

Additional comments

In this retrospective study you should find and mention the specifity and sensitivity of the cut-off values. why you did not use ROC analyses?
Why the results are not consistent with the literature? exempale: a prognostic factor must be in relation with other prognostic factor like TNM,...

Reviewer 3 ·

Basic reporting

Some grammatical errors exist in the manu.Please exam carefully and correct.

Experimental design

1.The cut-off value of MPV/PC ratio was obtained by X-tile plot.Please compare the method X-tile plot and ROC curve.
2.Please give more details in Figure 4.

Validity of the findings

The prognosis of ESCC is determined by many factors. The inflammation is important in the development of cancer.Please analysis the impact of neutrophil/ platelet ratio on prognosis and discuss it.

Additional comments

The author invested the prognostic role of MPV/PC ratio in ESCC patients which was an interesting topic.
A low MPV/PC ratio level (≤0.04) was associated with poor CSS in ESCC.But we all konw, the prognosis of ESCC is determined by many factors. The inflammation is important in the development of cancer.Please analysis the impact of neutrophil/ platelet ratio on prognosis and discuss it.Besides,.The cut-off value of MPV/PC ratio was obtained by X-tile plot.Please compare the method X-tile plot and ROC curve.Some grammatical errors exist in the manu.Please exam carefully and correct.

Reviewer 4 ·

Basic reporting

Your introduction needs some improvements.
1.In line 62, what's the 8th incidence for EC, is it in China? or all over the world?
2.Please explain why you introduced MPV,PC and cardiovascular disease, in this study about esophageal cancer, in line 69-71.
3.In line74 "However, the role for MPV/PC ratio in ESCC has not yet been evaluated." Actually, there were several studies about MPV, COP-MPV and MPV/PC and EC, as you cited in the discussion part line 156-168. And also another study : Reduced mean platelet volume is associated with poor prognosis in esophageal cancer. Shen W, Cui MM, Wang X, Wang RT. Cancer Biomark. 2018;22(3):559-563. doi: 10.3233/CBM-181231.
So, what's the novelty of this study need more detail discussion.

Experimental design

The MPV (PC) and CRP detection methods need more sufficient details. What about the CRP detection? Is it serum CRP or whole blood CRP? And how about the detailed methods.

Validity of the findings

The conclusions should be supported by the data. But the novelty of this study need more detail discussion.

Additional comments

There are some grammar issues should be improved , for example lines 87, and so on.

---

## Round 0.2 · accepted · Accept

Many thanks for your support of PeerJ! Congratulations!

One minor change is recommended for the abstract. See attached.

Reviewer 1 ·

Basic reporting

This is a retrospective study that can clearly illustrate the prognostic value of MPV/PC for esophageal squamous cell carcinoma. However, for this study, the sample size is not large enough, and the selected statistical methods appear still questionable.
It has been addressed.

Experimental design

1.Did the researchers set up a validation group to verify the conclusion?
It has been addressed.

2.Line 121-122, the levels of MPV/PC ratio were significantly correlated with the CRP levels (P =0.029). CRP is an inflammatory marker just like MPV/PC. These two markers would interact with each other in the Cox regression model to influence the final conclusion.
It has been addressed.

Validity of the findings

1.CEA is closely related to the occurrence, recurrence and metastasis of ESCC. Why didn’t researchers consider CEA as one of the prognostic factors of CSS?
It has been addressed.

2. Why did the researchers use the X-tile program instead of the ROC to calculate the optimal cut-off values? What are the advantages of this method relative to the ROC?
It has been addressed.

3. There are some grammatical errors/writing style issues. e.g in the abstract, “respectively” is needed at the end of the first sentence in the results section.
It has been addressed.

Reviewer 4 ·

Basic reporting

no comment

Experimental design

no comment

Validity of the findings

no comment

Additional comments

no comment